# Impact of the COVID-19 Pandemic on the Level of Anxiety and Depression in Caregivers of Children Benefiting from Neurorehabilitation Services

**DOI:** 10.3390/ijerph20054564

**Published:** 2023-03-04

**Authors:** Lidia Perenc, Justyna Podgórska-Bednarz, Agnieszka Guzik, Mariusz Drużbicki

**Affiliations:** Institute of Health Sciences, Medical College, University of Rzeszow, 35-310 Rzeszow, Poland

**Keywords:** COVID-19 pandemic, anxiety and depression, child caregivers

## Abstract

Introduction: At the turn of March and April 2020, due to the occurrence of COVID-19 in Poland, the first restrictions on the provision of rehabilitation services were introduced. Nevertheless, caregivers strived to ensure that their children could benefit from rehabilitation services. Aim of the study: To determine which of the selected data presented in the media reflecting the intensity of the COVID-19 epidemic in Poland differentiated the level of anxiety and depression in caregivers of children benefiting from neurorehabilitation services. Material and methods: The study group consisted of caregivers of children (*n* = 454) receiving various neurorehabilitation services in the inpatient ward of Neurological Rehabilitation of Children and Adolescents (*n* = 200, 44%), in the Neurorehabilitation Day Ward (*n* = 168, 37%), and in the Outpatient Clinic (*n* = 86, 19%) of the Clinical Regional Rehabilitation and Education Center in Rzeszow. The average age of the respondents was 37.23 ± 7.14 years. The Hospital Anxiety and Depression Scale (HADS) was used to measure the severity of anxiety and depression in caregivers of children. The questionnaires were distributed from June 2020 to April 2021. As a measure of the severity of the COVID-19 epidemic in Poland, the figures presented in the media were adopted. In addition, data on the COVID-19 pandemic presented in the media (Wikipedia, TVP Info, Polsat Nes, Radio Zet) on the day preceding the completion of the survey were analyzed based on statistical analysis methods. Results: 73 of the surveyed caregivers (16.08%) suffered from severe anxiety disorders, and 21 (4.63%) from severe depressive disorders. The average severity of anxiety (HADS) in the subjects was 6.37 points, and the average severity of depression was 4.09 points. There was no statistically significant relationship between the data presented in the media—such as daily number of infections, total number of infections, daily number of deaths, total number of deaths, total number of recoveries, number of hospitalizations, and people under quarantine—and the level of anxiety and depression of the studied caregivers (*p* > 0.05). Conclusions: It was not found that the selected data presented in the media, showing the intensity of the COVID-19 epidemic in Poland, significantly differentiated the level of anxiety and depression among caregivers of children using neurorehabilitation services. Their motivation to continue the treatment, caused by concern for their children’s health, resulted in less severe symptoms of anxiety and depression during the peak period of the COVID-19 pandemic.

## 1. Introduction

The media (Internet, TV, radio) contained information about the actual situation and spread of the coronavirus disease (COVID-19) [1,2,3]. It was feared that simply providing updates by the government about the epidemiological situation during the COVID-19 pandemic in the media might not be enough to draw public attention to the threat [4]. It was shown that during the COVID-19 pandemic, anxiety and depression were common problems in the general population [5,6], as well as in people suffering from COVID-19 [7], and mental health problems worsened during isolation [8]. As is known, anxiety and depression can occur as isolated disorders or coexist [9]. Various posts in social media reflected the interest and emotions evoked by the COVID-19 pandemic [10]. Spontaneous searching for information on the COVID-19 pandemic resulted in an increase in the amount of time spent browsing the Internet, watching TV, listening to the radio, and looking at newspapers, and was associated with unpleasant mental experiences [11].

Much has been written in the literature about the impact of the COVID-19 pandemic on the general population, while little attention has been paid to the difficulties faced by children with neurological disorders and their caregivers. Compared to the general population, this group had to struggle to secure the necessary medical care and therapy because, due to the clinical characteristics of these disorders, disabled children required continuous rehabilitative treatment. It was often very difficult because during the intensification of the pandemic many rehabilitation centers were closed, which deprived children with disabilities from access to rehabilitation services. For children with neurological disorders, this led to negative consequences such as increased muscle tone/spasticity, pain, poorly fitting orthotics, and delay in getting therapeutic equipment. In addition, caregivers were impacted as they had to perform more of the therapeutic exercises with their children at home [12].

The interruption of rehabilitation services due to a COVID-19 pandemic-related lockdown can significantly impact the functional abilities of patients with chronic neurological diseases [13]. In children, the most common neurodevelopmental disorders are cerebral palsy, myelomeningocele, hydrocephalus, spinal muscle atrophy, congenital polyneuropathy, muscle dystrophy, and autism spectrum disorders [14,15]. The COVID-19 pandemic caused prolonged home isolation and forced caregivers of children with neurodevelopmental disorders to change their daily strategies to prevent their children from getting worse. It was necessary to introduce regular online consultations, conduct online therapy, educate the child in the field of COVID-19 and preventive behaviors, create a structured daily schedule and a reinforcement system, and select activities appropriate for the child [15]. Caregivers of children with special needs assessed performing independent therapeutic tasks at home during the COVID-19 pandemic as difficult [16]. A number of similar demands related to the care of autistic children during the COVID-19 pandemic increased the risk of mental health problems for caregivers [17]. Solutions were sought for this difficult situation related to the pandemic. For example, given the difficulties of COVID-19, a novel, virtual, multidisciplinary, short intervention program for families with children with neurodevelopmental disorders was developed and rapidly implemented at the Hospital for Sick Children (SickKids) in Toronto, Ontario, Canada [18].

The COVID-19 pandemic also posed new organizational challenges to the Polish healthcare system [19]. Changes were gradually introduced in the healthcare units, which had to reconcile the satisfaction of the health needs of patients and the needs of medical workers related to ensuring safety [20]. At the turn of March and April 2020, due to the occurrence of COVID-19 in Poland, the first restrictions on the provision of various rehabilitation services were introduced [21]. This resulted in the termination of the work of the units in which the tests were carried out for less than two months. This also applied to centers operating in our region. In April 2020, the Clinical Regional Rehabilitation and Education Center in Rzeszow served as an isolation facility [22]. As a result, caregivers faced a serious problem because their children were deprived of access to neurorehabilitation. After resuming work, caregivers strived to ensure that their children could benefit from rehabilitation services. Surprisingly, they expressed no significant concerns about the pandemic.

In many of them, an increase in the intensity of symptoms such as anxiety, depression, or a sense of helplessness was observed. Results of similar research show that perceived treatment control over the illness and its course is related to mild to severe symptoms of anxiety and depression among parents of children with neurological disorders [23]. It has also been shown that emotion- and avoidance-oriented coping styles and socioeconomic status are crucial factors in the adjustment process of parents of children with neurodevelopmental conditions. By contrast, parenting stress and child difficulties were the most significant predictors of negative psychological outcomes in such parents [24].

As is known, parents’ attitudes and psychological adjustment during their child’s implementation of the neurorehabilitation program are key aspects for the child’s adherence to care and the impact of the disease. Therefore, it can be expected that the conditions caused by the COVID-19 pandemic will contribute to the occurrence of negative psychological consequences in them, which may disrupt the course of pediatric neurorehabilitation.

The aim of this study was to determine which of the selected data presented in the media showing the intensity of the COVID-19 pandemic differentiated the level of anxiety and depression in caregivers of children using neurorehabilitation services. Another research aim was identification of predictors that might help professionals to develop screening procedures to identify parents (caregivers) at high risk for anxiety or depression. Knowing these predictors would enable them to conduct early interventions to reduce uncertainty and maladaptive coping strategies that may influence neurorehabilitation processes.

## 2. Material and Methods

### 2.1. Participants

The study group consisted of caregivers of children (*n* = 454) using various neurorehabilitation services: in the Inpatient Department of Neurological Rehabilitation of Children and Adolescents (44.05%), in the Day Department of Neurorehabilitation (37%), and in the Outpatient Rehabilitation Clinic for Children and Adolescents (19%) of the Clinical Regional Rehabilitation and Education Center in Rzeszow. The mean age of the subjects was 37.23 ± 7.14 years. During their stay in a rehabilitation center, patients in the developmental age were most often under the care of women (83.04%), and less frequently by men (16.52%) or non-binary persons (0.44%). In 30.40% of cases there were caregivers of children with a disability certificate (receiving a care allowance). Caregivers slightly more often came from a rural environment (52.86%) than from an urban one (47.14%). In terms of religion, 33.92% of cases indicated that they were strongly religiously involved, 58.81% indicated that they were averagely involved, 6.83% indicated that they were indifferent to religion, and in 0.44% religion raised objections in them. The caregivers most often had higher education (49.56%), slightly less often secondary education (48.46%), and extremely rarely primary education (1.98%). In 5.95% of cases, admission to the center was sudden and unexpected, and in 94.05% it was planned. Chronic ailments were the reason for admission to a rehabilitation center for 89.65% of cases, and 10.35% for those that occurred suddenly. A total of 46.92% of the respondents stayed in the center in summer, 24.89% in autumn, 22.25% in winter, and 5.95% in spring. A caregiver with a negative PCR test for coronavirus infection (performed once immediately before the child was admitted to the ward) could stay in the inpatient ward with the child (55.95% of cases). In the day ward and outpatient care, a PCR test for coronavirus infection was not performed for caregivers (44.05% of cases) (Table 1).

### 2.2. Procedures and Data Analysis

The main tool used in this study was a questionnaire. In the introductory part, questions about the date of the study, demographics, and other data were included, on the basis of which the above characteristics of the study group were presented. Attention was paid to whether the respondent cared for a child with a certified disability, because then he or she is entitled to a care allowance. In addition, it was considered whether the child’s caregiver was in an emergency situation, or whether the child’s ailments occurred suddenly. Not all children with ailments appearing suddenly are admitted urgently; some have a scheduled, short-term admission date. Nursing staff assist in the care of children admitted to the stationary ward. The Hospital Anxiety and Depression Scale (HADS) was used to measure the severity of anxiety (Cronbach’s alpha = 0.83) and depression (Cronbach’s alpha = 0.82) [25] in caregivers of children. The HADS results were interpreted according to the key where a score from 0 to 7 points means no disturbances, a score from 8 to 10 points means a borderline state, and a score from 11 to 21 points means significant disturbances. The questionnaires were distributed from 30 June 2020 to 20 April 2021 in direct contact. Parents were asked whether they listened to or read news about the severity of the COVID-19 pandemic in the media the day before completing the survey. The whole group of surveyed parents confirmed that they listened to or read news about the severity of the COVID-19 pandemic in the media the day before completing the survey (100%, *n* = 454). The confirmation of such behavior was necessary to qualify the participants for the rest of the study. The participants were informed about the method of completing the questionnaire, and questions were answered as to the circumstances of the research and the method of completing the questionnaire, as well as the places where it could be submitted anonymously. Due to the negative coronavirus infection status of the caregivers of children staying in the inpatient ward and the unknown status of those from the day ward and outpatient clinic, two appropriate places for submitting questionnaires were organized. The data presented in the media (which was checked on Wikipedia for consistency with those provided in Poland on television, radio, and government social media [26], TVP Info [27], Polsat News [28], Radio Zet [29]) on the COVID-19 pandemic from the day preceding the completion of the survey were analyzed, i.e., daily number of infections, total number of infections, daily number of deaths, total number of deaths, total number of recoveries, number of hospitalizations, and people quarantined. These data were considered to reflect the intensity of the pandemic. Most of the data were established on the basis of tables presented in Wikipedia; single missing data were supplemented from other sources, such as TVP Info, Polsat News, and Radio Zet. The search engine entered the phrase ‘coronavirus in Poland’, date (day, month, year) and the name of the media (TVP Info, Polsat News, or Radio Zet).

The analysis of quantitative variables was carried out by calculating the mean, standard deviation, median, and quartiles. The analysis of qualitative variables was carried out by calculating the number and percentage of occurrences of each value. The comparison of qualitative variables in the two groups was performed using the Mann–Whitney test. In case of qualitative variables, the comparison in three or more groups was performed using the Kruskal–Wallis test. After detecting statistically significant differences, a post-hoc analysis was performed using Dunn’s test to identify statistically significant groups. Correlations between quantitative variables were analyzed using Spearman’s rank correlation coefficient. Multivariate analysis of the influence of many variables on the quantitative variable was performed using the linear regression method. The results are presented as regression model parameter values with a 95% confidence interval. Thus, all *p* values below 0.05 were interpreted as significant associations. Single- and multi-factor analysis of the influence of many variables on a binary variable was performed using the logistic regression method. The results are presented as OR parameter values with a 95% confidence interval. A significance level of 0.05 was adopted in all analyses. When the absolute values of Spearman’s rank correlation coefficient (R) were lower than 0.2, this indicated lack of linear relationship. The analysis was performed with use of the R program, version 4.1.0 [30].

Significant female dominance was observed in the study group (*n* = 377, 83.04%). However, the tests used in statistical analysis (Mann–Whitney test, Kruskal–Wallis test, and multivariate analysis) minimize the influence of unequal subgroups of the study group. In these tests, results indicative of statistical significance are the harder to obtain, since the large inequality of subgroups is noticed; therefore, the results are adequate because the tests are designed to take the inequalities into consideration.

In all analyses, caregivers who were indifferent or opposed to religion were combined into one subgroup. In the multivariate analysis, the caregiver’s coronavirus infection status marked on admission to the center was omitted. This factor overlaps to a large extent with the mode of admission to the center. All caregivers admitted to the inpatient ward had a negative infection status, and in those admitted to the outpatient or day ward this status was unknown.

The study was approved by the Bioethics Committee of the University of Rzeszow (resolution no. 3/07/2020). Before submitting the application to the Bioethics Committee of the University of Rzeszow, all required consents for the study were obtained (17 June 2020).

## 3. Results

### 3.1. Statistical Characteristics of the Intensity of the COVID-19 Pandemic in Poland

In the analyzed period, the intensity of the COVID-19 pandemic varied greatly. Here are examples of minimum and maximum value ranges of the analyzed feature: daily number of infections: 0.24–34.15 (thousands), total number of infections: 34.15–2695.33 (thousands), daily number of deaths: 1–954, total number of deaths: 1.45–62.13 (thousands), total number of recoveries: 20.9–2334.98 (thousands), number of hospitalizations: 1.56–34.86 (thousands), and people quarantined: 72.28–489.08 (thousands) (Table 2).

### 3.2. Statistical Characteristics of Anxiety and Depression Intensity, Occurrence of Anxiety and Depression, and Their Coexistence in the Study Group

The average intensity of anxiety (HADS) in the subjects was 6.37 points, standard deviation 3.88. The intensity of anxiety ranged from 0 to 19 points. The mean intensity of depression (HADS) was 4.09 points, the standard deviation was 3.22, and the range of values was between 0 and 18 points (Table 3).

In the case of anxiety, 294 out of 454 survey participants (64.76%) had no visible symptoms of disorders, 87 respondents (19.16%) presented a borderline state, and 73 respondents (16.08%) showed clear disorders. In the case of depression, 393 out of 454 survey participants (86.56%) had no disorders, 40 respondents (8.81%) presented a borderline state, and 21 respondents (4.63%) had clear symptoms of disorders (see ‘A’ in Table 4). In the further part of the work, clear disorders in the case of anxiety will be referred to as anxiety, and similarly clear disorders in the case of depression as depression.

Clear symptoms of anxiety and depression coexisted in 19 surveyed caregivers (4.19%) (see ‘B’ in Table 4). The higher the level of anxiety, the higher the level of depression; and the higher the level of depression, the higher the level of anxiety. The presented relationship is statistically significant: *p* < 0.001 (Table 5).

### 3.3. Selected Data on the Epidemic Presented in the Media and the Level of Anxiety and Depression: Univariate Analysis

Selected data on the epidemic presented in the media did not differentiate the level of anxiety. According to the statistical analysis performed with the usage of the Spearman rank correlation, no statistically significant correlation was found (Table 6). For different correlations between specific parameters reflecting the severity of the pandemic and the level of anxiety, the following *p* values (all >0.05) were obtained: (A) daily number of infections: *p* = 0.992, (B) total number of infections: *p* = 0.995, (C) daily number of deaths: *p* = 0.834, (D) total number of deaths: *p* = 0.995, (E) total number of recoveries: *p* = 0.995, (F) number of hospitalizations: *p* = 0.853, and (G) daily number of people under quarantine: *p* = 0.997. 

Similarly, selected data on the epidemic presented in the media did not differentiate the level of depression. In this case, according to the statistical analysis performed with the usage of the Spearman rank correlation, no statistically significant correlation was found (Table 6). For different correlations between specific parameters reflecting the severity of the pandemic and the level of depression, the following *p* values (all > 0.05) were obtained: (A) daily number of infections: *p* = 0.968, (B) total number of infections: *p* = 0.778, (C) daily number of deaths: *p* = 0.502, (D) total number of deaths: *p* = 0.778, (E) total number of recoveries: *p* = 0.778, (F) number of hospitalizations: *p* = 0.875, and (G) daily number of people under quarantine: *p* = 0.781. 

Furthermore, the absolute values of the Spearman rank correlation coefficient (R) in every case were lower than 0.2, which indicates lack of linear relationship (Table 6). For specific correlations, the following values of R were obtained: (A) daily number of infections and the level of anxiety: R = 0, daily number of infections and the level of depression: R = −0.002; (B) total number of infections and the level of anxiety: R = 0, total number of infections and the level of depression: R = −0.013; (C) daily number of deaths and the level of anxiety: R = 0.01, daily number of deaths and the level of depression: R = −0.032; (D) total number of deaths and the level of anxiety: R = 0, total number of deaths and the level of depression: R = −0.013; (E) total number of recoveries and the level of anxiety: R = 0, total number of recoveries and the level of depression: R = −0.013; (F) number of hospitalizations and the level of anxiety: R = 0.009, number of hospitalizations and the level of depression: R = 0.007; and (G) daily number of people under quarantine and the level of anxiety: R = 0, daily number of people under quarantine and the level of depression: R = 0.013.

### 3.4. Selected Parameters Characterizing the Study Group and the Level of Anxiety and Depression: Univariate Analysis

Based on univariate analysis, it was found that neither age, gender, status of the caregiver, nor his or her social environment differentiated the level of anxiety and depression (see ‘A–D’ in Table 7). The caregivers who were indifferent or opposed to religion were combined into one subgroup (see ‘A’ in Table 8). This was similar in the case of the caregiver’s religious involvement, emergency/planned hospital admission, mode of admission of the child to the center (see ‘A’,’C–D’ in Table 8), season of the year during which the stay in the center took place, and infection status (coronavirus) of the caregiver marked upon admission to the center (see ‘B’,’C’ in Table 9). The *p* values for these relationships do not meet the condition of assumed statistical significance and are greater than 0.05.

Statistically significant relationships were also obtained. The lower the level of education of the caregiver, the higher the level of depression. The level of depression in caregivers with higher education was 3.68 points, with secondary education 4.37 points, and with primary education it was the highest at 7.56 points. The *p* values for this relationship indicated statistical significance: *p* < 0.001 (see ‘B’ in Table 8). It was also found that in the case of admission to the center due to ailments occurring in the child in a sudden and unexpected way, the caregiver’s level of depression was higher and amounted to 5.4 points; and due to persistent ailments, it was lower and reached 3.94 points. This relationship was statistically significant: *p* = 0.01 (see ‘A’ in Table 9).

### 3.5. Predictors Differentiating the Level of Anxiety: Multivariate Linear Regression

The R² coefficient for this model was 7.56%, which means that 7.56% of the variability in the level of anxiety was explained by the variables included in the model. The remaining 92.44% depends on variables not included in the model and random factors. Based on the multivariate linear regression model, it was shown that statistically significant (*p* < 0.05) and independent predictors of the level of anxiety are (Table 10):Female gender (for females, the regression parameter was 1.019, so women had an average level of anxiety 1.019 points higher than men (*p* = 0.041)).Indifference or lack of religious commitment (the regression parameter was 1.699, so those with indifference or lack of religious commitment had an average level of anxiety 1.699 points higher compared to those with strong religious commitment (*p* = 0.026)).Admission to an inpatient ward (the regression parameter was −1.267, so caregivers whose children were admitted to an inpatient ward had an average level of anxiety 1.267 points lower than those whose children were admitted to an outpatient department, *p* = 0.015)).

### 3.6. Predictors Differentiating the Level of Depression: Multivariate Linear Regression

The R² coefficient for this model was 10.29%, which means that 10.29% of the variability in the level of depression was explained by the variables included in the model. The remaining 89.71% depends on variables not included in the model and random factors. Based on the multivariate linear regression model, it was shown that statistically significant (*p* < 0.05) and independent predictors of the level of depression are (Table 11):Secondary education of the caregiver (the regression parameter was 0.697, so in caregivers with secondary education the level of depression was higher by 0.697 points on average compared to caregivers with higher education; *p* = 0.026).Primary education of the caregiver (the regression parameter was 3.832, so in caregivers with primary education the level of depression was higher on average by 3.832 points compared to caregivers with higher education; *p* = 0.001).The child’s ailments that occurred suddenly as the reason for admission to the hospital (the regression parameter was 1.636, so the level of depression among caregivers of children admitted to the hospital due to emergency ailments was higher by 1.636 points on average compared to caregivers of children with chronic ailments; *p* = 0.009).

Multivariate linear regression tests were performed in order to determine factors significantly influencing the level of depression. The results reflecting the level of depression in caregivers with higher levels of education were defined to be a reference point for this analysis. It was seen that the lower the level of education of a caregiver, the higher their level of depression assessed on the HADS scale. Higher level of caregiver education can be considered a protective factor in terms of depression, as it is correlated with the lowest level of depression assessed with the HADS scale. An elementary level of caregiver education can be considered a risk factor for depression, because it is correlated with the highest level of depression according to the HADS scale.

### 3.7. Predictors of the Co-Occurrence of Anxiety and Depression: Univariate Analysis

Based on the results of research using logistic regression models, carried out separately for each analyzed feature, it was found that statistically significant (*p* < 0.05) predictors of the chance of co-occurrence of anxiety and depression are (Table 12):Secondary education: for this parameter, the odds ratio is equal to 0.115, so secondary education reduced the odds of co-occurrence anxiety and depression by 88.5% compared to primary education (*p* = 0.015).Higher education: for this parameter, the odds ratio is equal to 0.163, so higher education lowered the odds of co-occurrence anxiety and depression by 83.7% compared to primary education (*p* = 0.036).Admission to the inpatient department of the center: for this parameter, the odds ratio is equal to 0.289, so admission to the inpatient department of the center reduced the odds of co-occurrence anxiety and depression by 71.1% compared to admission to the outpatient department (*p* = 0.039).

Based on the univariate analysis, it was shown that the caregiver’s secondary and higher education lowers the risk of co-occurring anxiety and depression in comparison to primary education. A higher level of education lowers the risk of co-occurrence of depression and anxiety more than average. In this analysis, the frequency of co-occurrence of depression and anxiety in people with primary education was defined as a reference point.

### 3.8. Predictors of the Co-Occurrence of Anxiety and Depression: Multivariate Analysis

Based on the results of studies using a multivariate logistic regression model, it was shown that the statistically significant (*p* < 0.05) and independent predictors of the chance of co-occurrence of anxiety and depression are (Table 13):Caregiver’s secondary education: for this parameter, the odds ratio is equal to 0.07, so secondary education reduced the odds of co-occurrence of anxiety and depression by 93.0% compared to primary education (*p* = 0.011).Caregiver’s higher education: for this parameter, the odds ratio is equal to 0.072, so higher education reduced the odds of co-occurrence of anxiety and depression by 92.8% compared to primary education (*p* = 0.012).Admission of the child to the inpatient ward of the center: admission to the inpatient ward lowered the odds of co-occurrence of anxiety and depression by 0.224, and thus it lowered the odds of co-occurring anxiety and depression by 77.6% compared to admission to the outpatient clinic (*p* = 0.029).

Anxiety co-occurred with depression in only 19 people. This raises the question of whether the above analysis—multivariate logistic regression—should be implemented for all variables. Multivariate logistic regression was repeated only for two variables for which a statistically significant relationship was obtained in the single-factor logistic regression model: caregiver education and the mode of admission to the health center. Such a model showed that significant (*p* < 0.05) independent predictors of the co-occurrence of anxiety and depression are (Table 14):Secondary education of the caregiver: for this parameter, the odds ratio is equal to 0.094, so secondary education reduces the chance of co-occurring anxiety and depression by 91.6% compared to primary education.Caregiver’s higher education: for this parameter, the odds ratio is equal to 0.115, so higher education reduces the chance of co-occurring anxiety and depression by 88.5% compared to primary education.Admission of a child to the inpatient ward in the health center: admission to the inpatient ward lowers the odds of co-occurring anxiety and depression to 0.261, so it reduces the odds of co-occurring anxiety and depression by 73.9% compared to admission to the outpatient clinic.

Based on the multivariate analysis, it was shown that the caregiver’s secondary and higher education lowers the risk of co-occurring anxiety and depression in comparison to primary education. A higher level of education lowers the risk of co-occurrence of depression and anxiety more than average. In this analysis of the frequency of co-occurrence of depression and anxiety, primary level of education was defined as a reference point.

## 4. Discussion

Among caregivers of children benefiting from neurorehabilitation services during the COVID-19 pandemic, the incidence of borderline state and clear symptoms of anxiety (HADS) was 35.68% and the incidence of borderline state and clear symptoms of depression (HADS) was 13.44%. Other researchers showed that among parents of children with neurological disorders, the incidence of similar anxiety states (HADS) was 41.0%, and similar depressive states (HADS) was 39.5% [22]. In another study conducted among caregivers of children with special needs in India, from 29 April 2020 to 22 May 2020, during the COVID-19 pandemic, the prevalence of symptoms of depression, anxiety, and stress was 62.5%, 20.5%, and 36.4%, respectively (Depression Anxiety Stress-21 Scale) [15]. On the basis of research conducted in Italy among parents of children with neuropsychiatric diagnosis, an increase in stress levels (Parenting Stress Index—Short Form) was found during the COVID-19 pandemic compared to the pre-pandemic period [31,32].

In Italy, there was also a rehabilitation lockdown during the COVID-19 pandemic. Research conducted in the period from 26 March 2020 to 11 May 2020 showed that caregivers of children with disabilities reported anxiety and depression symptoms and concerns about their child’s development during the rehabilitation lockdown, but the main concerns were also related to the risk of COVID-19 contagion [33]. Meanwhile, Chinese researchers found that the alarming data provided by the government on the epidemiological situation during the COVID-19 pandemic were not interpreted by the general population as a state of emergency [4]. This is a similar tendency as in our study, where selected epidemic data presented in the media—daily number of infections, total number of infections, daily number of deaths, total number of deaths, total number of recoveries, number of hospitalizations, and daily number of people quarantined—did not differentiate levels of anxiety and depression. 

It has been shown that female gender, admission to an inpatient ward, and indifference or lack of religious commitment are statistically significant and independent predictors of the level of anxiety. They account for 7.56% of the variability in the level of anxiety in the adopted multivariate linear regression model. Female caregivers had a higher level of anxiety than male caregivers. A decreased level of anxiety was present in caregivers whose children were admitted to the inpatient ward compared to those whose children were admitted to the outpatient clinic. An increased level of anxiety was reported by caregivers showing indifference or lack of religious commitment compared to those showing strong religious commitment. In this study, female caregivers predominated. The statistical tests used in our study abolish the effect of inequality in the numbers of subgroups. Therefore, it cannot be said that female sex is a predictor of the level of anxiety, because more women participated in the study. It was noted that a female predominance has also been observed in other studies on the prevalence of anxiety and depression during the COVID-19 pandemic in the general population [5]. In our rehabilitation center, the closeness of the staff is most prevalent in the inpatient ward. Caregivers of children requiring neurorehabilitation admitted to this unit had the lowest level of anxiety. In the opinion of caregivers of patients with neuromuscular diseases, the constant support of the center and the proximity of the center staff was a positive aspect of the healthcare services provided in the rehabilitation center during the COVID-19 pandemic [12]. In another study conducted among parents of children with neurological diseases, other predictors were found to differentiate the level of anxiety (HADS): perceived treatment control over the illness, perceived understanding the illness, and perceived personal control over the illness (Brief Illness Perception Questionnaire) [23]. 

It has been shown that statistically significant and independent predictors of the level of depression are the caregiver’s secondary education, the caregiver’s primary education, and the child’s ailments that occurred suddenly as the reason for admission to the hospital. These predictors account for 10.29% of the variability in the level of depression in the surveyed caregivers in the multivariate linear regression model. Caregivers with primary education and caregivers with secondary education had higher levels of depression compared to caregivers with higher education. Caregivers of children with sudden health decline being the reason for admission to the hospital were characterized by higher level of depression compared to caregivers of children with chronic ailments. Difficulties in access to medical care caused by the COVID-19 pandemic affected the caregivers of patients with spinal muscular atrophy. During the COVID-19 pandemic, caregivers were required to make decisions about the treatment and urgent interventions included, while taking the risks associated with the severity of symptoms and the threat of the coronavirus pandemic into consideration. It was a new challenge, unknown prior to the pandemic [34]. In another study conducted among parents of children with neurological diseases, other statistically significant predictors differentiating the level of depression (HADS) were found: perceived treatment control over the illness, perceived understanding of the illness, and perceived timeline of the illness (Brief Illness Perception Questionnaire) [23].

Anxiety and depression (marked disorders) coexisted in 4.19%. It was found that there is a statistically significant relationship between the level of anxiety and depression: the higher the level of anxiety, the higher the level of depression; and the higher the level of depression, the higher the level of anxiety. It was found that statistically significant predictors reducing the probability of co-occurrence of anxiety and depression are secondary and higher education, and admission to the inpatient department of the center. Other researchers showed that severe depression in parents of children with neurodevelopmental disorders in May 2020 had a synergistic effect on severe parental stress and severe depressive state in May 2021 [35].

Our study showed that caregivers of children undergoing neurorehabilitation treatments showed high motivation to continue them. It is likely that this motivation, caused by concern for their children’s health, resulted in less severe symptoms of anxiety and depression during the peak period of the COVID-19 pandemic. This seems to be one of the most important findings from our research.

## 5. Conclusions

None of the selected data presented in the media showing the severity of the COVID-19 pandemic differentiated the level of anxiety and depression among caregivers of children using neurorehabilitation services.

It has been shown that significant predictors increasing the level of anxiety of caregivers of children using neurorehabilitation services during the COVID-19 pandemic are female compared to male gender, and indifference or opposition to religious commitment compared to strong religious commitment.

It was also shown that a significant predictor increasing the level of anxiety of caregivers of children using neurorehabilitation services during the COVID-19 pandemic was the child’s admission to an inpatient ward compared to the child’s admission to an outpatient clinic.

It has been shown that significant predictors increasing the level of depression in caregivers of children using neurorehabilitation services during the COVID-19 pandemic are secondary and primary education of the caregiver compared to higher education, and ailments of the child that occurred suddenly as the reason for hospitalization compared to chronic ailments.

Additionally, it has been shown that significant predictors reducing the co-occurrence of anxiety and depression in caregivers of children using neurorehabilitation services during the COVID-19 pandemic are secondary and higher education of the caregiver compared to primary education, and admission of a child to an inpatient ward compared to an outpatient department.

a.Defining the direction of future research

Exploring more of the factors that shape anxiety and depression levels in caregivers of children, targeting their children, spouses, work environment, and more. 

b.Limitations

Not taking into account more factors shaping the level of anxiety and depression in caregivers of children, targeting their children, spouses, work environment, and others.

## Figures and Tables

**Table 1 ijerph-20-04564-t001:** Characteristics of the studied group.

Parameter Characterizing the Studied Group	Numerical/Percentage Characteristics	
Age of caregiver (years)	Mean ± standard deviation	37.23 ± 7.14
	Median,	37
	lower and upper quartile	32–42
Caregiver’s gender	Male	75 (16.52%)
	Female	377 (83.04%)
	Non-binary person	2 (0.44%)
Caregiver’s status	Caregiver of a disabled child	138 (30.40%)
	Caregiver of a child	316 (69.60%)
Caregiver’s social environment	Rural	240 (52.86%)
	Urban	214 (47.14%)
Religious involvement of the caregiver	Strong	154 (33.92%)
	Medium	267 (58.81%)
	Neutral	31 (6.83%)
	Strongly opposed	2 (0.44%)
Caregiver education	Elementary	9 (1.98%)
	Secondary	220 (48.46%)
	Higher	225 (49.56%)
Admission of the child to the center	Urgent	27 (5.95%)
	Planned	427 (94.05%)
Mode of admitting a child to the center	Outpatient clinic	86 (18.94%)
	Day ward	168 (37.00%)
	Inpatient ward	200 (44.05%)
Reason for admitting the child to the center	Chronic ailments	407 (89.65%)
	Ailments that occurred suddenly	47 (10.35%)
Season of stay at the center	Summer	213 (46.92%)
	Autumn	113 (24.89%)
	Winter	101 (22.25%)
	Spring	27 (5.95%)
Infection status (coronavirus) of the caregiver marked upon admission to the center	Unknown	254 (55.95%)
	Negative	200 (54.05%)

**Table 2 ijerph-20-04564-t002:** Statistical characteristics of the intensity of the COVID-19 pandemic in Poland in the analyzed period.

COVID-19 Pandemic in Poland from 29 June 2020 to 19 April 2021	Quantitative Characteristic	
Daily number of infections (thousands)	Mean ± standard deviation	5.82 (7.16)
	Median (lower and upper quartile)	0.9 (0.58–8.78)
	Range	0.24–34.15
Total number of infections (thousands)	Mean ± standard deviation	650.57 (793.25)
	Median (lower and upper quartile)	80.7 (57.28–1412)
	Range	34.15–2695.33
Daily number of deaths	Mean ± standard deviation	132.16 (187.66)
	Median (lower and upper quartile)	18 (11–241)
	Range	1–954
Total number of deaths (thousands)	Mean ± standard deviation	15.26 (18.74)
	Median (lower and upper quartile)	2.32 (1.89–32.36)
	Range	1.45–62.13
Total number of recoveries (thousands)	Mean ± standard deviation	511.37 (665.63)
	Median (lower and upper quartile)	64.97 (39.36–1153.65)
	Range	20.9–2334.98
Number of hospitalizations (thousands)	Mean ± standard deviation	9.16 (9.16)
	Median (lower and upper quartile)	2.22 (1.99–16.25)
	Range	1.56–34.86
Number of people quarantined (thousands)	Mean ± standard deviation	171.89 (100.99)
	Median (lower and upper quartile)	116.87 (101.05–206.05)
	Range	72.28–489.08

**Table 3 ijerph-20-04564-t003:** Statistical characteristics of the intensity of anxiety and depression in the respondents.

HADS (Points)	Mean	Standard Deviation	Median	Minimum	Maximum	Lower Quartile	Upper Quartile
Anxiety	6.37	3.88	6	0	19	4	9
Depression	4.09	3.22	3.5	0	18	2	6

**Table 4 ijerph-20-04564-t004:** Incidence of anxiety and depression disorders (A), incidence of coexistence of anxiety and depression (B).

Intensity	A. Incidence of Anxiety and Depression Disorders (HADS)	
	Anxiety	Depression
Lack of disorders	294 (64.76%)	393 (86.56%)
Borderline state	87 (19.16%)	40 (8.81%)
Pronounced disorders	73 (16.08%)	21 (4.63%)
B. Coexistence of anxiety and depression (pronounced disorders)		Incidence: 19 (4.19%)

**Table 5 ijerph-20-04564-t005:** Correlation between the level of anxiety and depression (Spearman rank correlation).

Examined Characteristics (HADS)	Spearman Rank Correlation (R)	Coefficient of Statistical Significance (*p*)
Level of anxiety (pts)/Level of depression (pts)	R = 0.742	*p* < 0.001 *

*—statistically significant relationship (*p* < 0.05).

**Table 6 ijerph-20-04564-t006:** Selected data on the epidemic presented in the media and the level of anxiety and depression: univariate analysis (Spearman’s rank correlation).

Level of Anxiety and Depression	Data on the Epidemic Presented in the MediaSpearman Rank Correlation (R), Coefficient of Statistical Significance (*p*)
HADS (points)	A. Daily number of infections
Anxiety	R = 0, *p* = 0.992
Depression	R = −0.002, *p* = 0.968
HADS (points)	B. Total number of infections
Anxiety	R = 0, *p* = 0.995
Depression	R = −0.013, *p* = 0.778
HADS (points)	C. Daily number of deaths
Anxiety	R = 0.01, *p* = 0.834
Depression	R = −0.032, *p* = 0.502
HADS (points)	D. Total number of deaths
Anxiety	R = 0, *p* = 0.995
Depression	R = −0.013, *p* = 0.778
HADS (points)	E. Total number of recoveries
Anxiety	R = 0, *p* = 0.995
Depression	R = −0.013, *p* = 0.778
HADS (points)	F. Daily number of hospitalization
Anxiety	R = 0.009, *p* = 0.853
Depression	R = 0.007, *p* = 0.875
HADS (points)	G. Daily number of quarantined persons
Anxiety	R = 0, *p* = 0.997
Depression	R = 0.013, *p* = 0.781

**Table 7 ijerph-20-04564-t007:** Age (A), gender of the caregiver (B), status of the caregiver (C), social environment (D) and the level of anxiety and depression.

HADS (points)		A. Age of Caregiver (Years)				
		Spearman Rank Correlation (R)				
Anxiety		R = −0.005, *p* = 0.922				
Depression		R = −0.018, *p* = 0.7				
HADS (points)		B. Caregiver’s gender (Kruskal–Wallis test)				*p*
		Male (*n* = 75)	Female (*n* = 377)		Non-binary person (*n* = 2)	
Anxiety	Mean ± SD	5.53 ± 3.62	6.52 ± 3.92		8 ± 2.83	*p* = 0.094
	Median	5	6		8	
	Quartile	3–7.5	4–9		7–9	
Depression	Mean ± SD	3.72 ± 3.29	4.16 ± 3.21		4.5 ± 0.71	*p* = 0.36
	Median	3	4		4.5	
	Quartile	1–5	2–6		4.25–4.75	
HADS (points)		C. Caregiver’s status (Mann–Whitney test)				*p*
		Caregiver of a child (*n* = 316)		Caregiver of a disabled child (*n* = 138)		
Anxiety	Mean ± SD	6.35 ± 3.89		6.41 ± 3.87		*p* = 0.826
	Median	6		6		
	Quartile	3.75–9		4–9		
Depression	Mean ± SD	3.97 ± 3.18		4.36 ± 3.28		*p* = 0.246
	Median	3		4		
	Kwartyle	1–6		2–7		
HADS (points)		D. Social environment (Mann–Whitney test)				*p*
		Rural (*n* = 240)		Urban (*n* = 214)		
Anxiety	Mean ± SD	6.44 ± 3.79		6.29 ± 3.99		*p* = 0.581
	Median	6		6		
	Quartile	4–9		3–9		
Depression	Mean ± SD	4.13 ± 3.14		4.04 ± 3.3		*p* = 0.541
	Median	4		3		
	Quartile	2–6		1–6		

*n*—number, SD—standard deviation, *p*—statistical significance coefficient.

**Table 8 ijerph-20-04564-t008:** Caregiver’s religious commitment (A), caregiver’s education (B), emergency/planned admission to the center (C), mode of admission to the center (D) and the level of anxiety and depression.

HADS (points)		A. Caregiver’s Religious Commitment (Kruskal–Wallis Test)				*p*
		Strong (*n* = 154)	Average (*n* = 267)		Indifferent or Strong Opposition (*n* = 33)	
Anxiety	Mean ± SD	6.16 ± 3.86	6.31 ± 3.79		7.79 ± 4.54	*p* = 0.168
	Median	6	6		8	
	Quartile	3 –8	4–9		4–11	
Depression	Mean ± SD	3.79 ± 3.21	4.18 ± 3.2		4.76 ± 3.35	*p* = 0.18
	Median	3	4		4	
	Quartile	1–6	2–6		2–7	
HADS (points)		B. Caregiver’s education (*p*: Kruskal–Wallis test + post-hoc analysis, Dunn’s test)				*p*
		Higher (*n* = 225)	Secondary (*n* = 220)		Elementary (*n* = 9)	
Anxiety	Mean ± SD	6.28 ± 3.72	6.36 ± 4		8.78 ± 4.35	*p* = 0.222
	Median	6	6		9	
	Quartile	4–8	3–9		6–10	
Depression	Mean ± SD	3.68 ± 3.14	4.37 ± 3.16		7.56 ± 3.81	*p* = 0.001 *
	Median	3	4		7	
	Quartile	1–5	2–7		5–10	C > B > A
HADS (points)		C. Child’s admission to the center (Mann–Whitney test)				*p*
		Emergency (*n* = 27)		Planned (*n* = 427)		
Anxiety	Mean ± SD	7.56 ± 4.07		6.29 ± 3.86		*p* = 0.1
	Median	7		6		
	Quartile	5.5–10		3.5–9		
Depression	Mean ± SD	5.07 ± 3.92		4.03 ± 3.16		*p* = 0.159
	Median	5		3		
	Quartile	2.5–7		1–6		
HADS (points)		D. Mode of child’s admission to the center (test Kruskal–Wallis)				*p*
		Outpatient ward (*n* = 86)	Day ward (*n* = 168)		Stationary ward (*n* = 200)	
Anxiety	Mean ± SD	6.98 ± 3.86	6.52 ± 4.22		5.97 ± 3.56	*p* = 0.189
	Median	6	6		6	
	Quartile	4–10	3–9		3–8	
Depression	Mean ± SD	4.19 ± 3.65	4.2 ± 3.23		3.95 ± 3.01	*p* = 0.779
	Median	3	4		3	
	Quartile	1–6.75	2–7		1.75–6	

*n*—number, SD—standard deviation, *p*—statistical significance coefficient, *—statistically significant relationship (*p* < 0.05).

**Table 9 ijerph-20-04564-t009:** Reason for admission to the center (A), season of the year during which the stay in the center took place (B), infection status (coronavirus) of the caregiver (C) and level of anxiety and depression marked upon admission to the center.

HADS (points)			A. Reason for admission of the child to the center (Mann–Whitney Test)				*p*
			Chronic Ailments (*n* = 407)		Sudden Ailments (*n* = 47)		
Anxiety	Mean ± SD		6.23 ± 3.78		7.53 ± 4.54		*p* = 0.057
	Median		6		7		
	Quartile		3–9		5–10		
Depression	Mean ± SD		3.94 ± 3.1		5.4 ± 3.88		*p* = 0.01 *
	Median		3		5		
	Quartile		1–6		2–7		
HADS (points)			B. Season of stay at the center (Kruskal–Wallis test)				*p*
			Summer (*n* = 213)	Autumn (*n* = 113)	Winter (*n* = 101)	Spring (*n* = 27)	
Anxiety		Mean ± SD	6.36 ± 3.81	6.14 ± 3.95	6.44 ± 4.01	7.07 ± 3.82	*p* = 0.763
		Median	6	6	6	6	
		Quartile	4–9	3–9	4–9	4–10	
Depression		Mean ± SD	4.04 ± 3.09	4.19 ± 3.26	3.94 ± 3.48	4.63 ± 3.12	*p* = 0.599
		Median	4	4	3	4	
		Quartile	2–6	1–7	1–6	2–7	
HADS (points)			C. Infection status (coronavirus) of the caregiver markedon admission to the center (Mann–Whitney test)				*p*
			Unknown (*n* = 254)		Negative (*n* = 200)		
Anxiety		Mean ± SD	6.68 ± 4.1		5.97 ± 3.56		*p* = 0.12
		Median	6		6		
		Quartile	4–9		3–8		
Depression		Mean ± SD	4.2 ± 3.37		3.95 ± 3.01		*p* = 0.29
		Median	4		3		
		Quartile	2–7		1.75–6		

*n*—number, SD—standard deviation, *p*—statistical significance coefficient, *—statistically significant relationship (*p* < 0.05).

**Table 10 ijerph-20-04564-t010:** Predictors differentiating the level of anxiety: multivariate analysis.

Characteristic		Parameter	95% CI		*p*
Caregiver’s gender	Male	ref.			
	Female	1.019	0.044	1.995	0.041 *
	Non-binary person	1.494	−4.215	7.203	0.608
Caregiver’s status	Caregiver of a child	ref.			
	Caregiver of a disabled child	0.255	−0.571	1.082	0.545
Caregiver’s social environment	Rural	ref.			
	Urban	−0.255	−1.004	0.493	0.505
Caregiver’s religious commitment	Strong	ref.			
	Moderate	0.068	−0.723	0.859	0.866
	Indifferent or opposed	1.699	0.207	3.191	0.026 *
Caregiver’s education	Higher	ref.			
	Secondary	0.154	−0.595	0.904	0.686
	Elementary	2.459	−0.231	5.148	0.074
Admission of the child to the center	Urgent	ref.			
	Planned	−0.399	−2.371	1.573	0.692
Mode of admitting a child to the center	Outpatient clinic	ref.			
	Day ward	−0.652	−1.675	0.372	0.213
	Inpatient ward	−1.267	−2.279	−0.254	0.015 *
Reason for admitting the child to the center	Chronic ailments	ref.			
	Ailments that occurred suddenly	1.344	−0.156	2.843	0.08
Caregiver’s age	(years)	0.006	−0.046	0.058	0.831
Season of stay at the center	Summer	ref.			
	Autumn	−0.347	−1.852	1.158	0.652
	Winter	−1.454	−4.819	1.911	0.397
	Spring	−3.066	−8.976	2.845	0.31
Daily number of infections	(thousands)	0.12	−0.131	0.371	0.348
Total number of infections	(thousands)	−0.014	−0.048	0.021	0.438
Daily number of deaths		−0.73	−5.363	3.902	0.757
Total number of deaths	(thousands)	0.117	−1.216	1.45	0.864
Total number of recoveries	(thousands)	0.012	−0.005	0.029	0.171
Number of hospitalizations	(thousands)	0.209	−0.664	1.081	0.639
Daily number of quarantined people	(thousands)	−0.006	−0.023	0.012	0.521

Multivariate linear regression, ref.—reference, CI—confidence interval, *p*—statistical significance coefficient, *—statistically significant relationship (*p* < 0.05).

**Table 11 ijerph-20-04564-t011:** Predictors differentiating the level of depression: multivariate analysis.

Characteristic		Parameter	95% CI		*p*
Caregiver’s gender	Male	ref.			
	Female	0.553	−0.242	1.349	0.174
	Non-binary person	−0.355	−5.01	4.3	0.881
Caregiver’s status	Caregiver of a child	ref.			
	Caregiver of a disabled child	0.319	−0.355	0.993	0.355
Caregiver’s social environment	Rural	ref.			
	Urban	−0.071	−0.681	0.539	0.82
Caregiver’s religious commitment	Strong	ref.			
	Moderate	0.283	−0.362	0.927	0.391
	Indifferent or opposed	0.794	−0.423	2.011	0.202
Caregiver’s education	Higher	ref.			
	Secondary	0.697	0.086	1.308	0.026 *
	Elementary	3.832	1.639	6.025	0.001 *
Admission of the child to the center	Urgent	ref.			
	Planned	0.168	−1.44	1.776	0.838
Mode of admitting a child to the center	Outpatient clinic	ref.			
	Day ward	−0.2	−1.034	0.635	0.64
	Inpatient ward	−0.56	−1.386	0.265	0.184
Reason for admitting the child to the center	Chronic ailments	ref.			
	Ailments that occurred suddenly	1.636	0.414	2.859	0.009 *
Caregiver’s age	(years)	0.002	−0.04	0.044	0.928
Season of stay at the center	Summer	ref.			
	Autumn	0.152	−1.075	1.379	0.809
	Winter	−1.759	−4.503	0.985	0.21
	Spring	−2.616	−7.436	2.203	0.288
Daily number of infections	(thousands)	−0.036	−0.24	0.169	0.732
Total number of infections	(thousands)	−0.028	−0.056	0	0.053
Daily number of deaths		0.719	−3.058	4.496	0.709
Total number of deaths	(thousands)	0.915	−0.172	2.002	0.1
Total number of recoveries	(thousands)	0.003	−0.011	0.017	0.654
Number of hospitalizations	(thousands)	0.514	−0.197	1.226	0.157
Daily number of quarantined people	(thousands)	−0.002	−0.016	0.012	0.778

Multivariate linear regression, ref.—reference, CI—confidence interval, *p*—statistical significance coefficient, *—statistically significant relationship (*p* < 0.05).

**Table 12 ijerph-20-04564-t012:** Predictors of the co-occurrence of anxiety and depression: univariate analysis.

Characteristic		Univariate Models			
		OR	95% CI		*p*
Caregiver’s gender	Male	1	ref.		
	Female	1.064	0.302	3.746	0.923
	Non-binary person	-	-	-	-
Caregiver’s status	Caregiver of a child	1	ref.		
	Caregiver of a disabled child	0.586	0.23	1.491	0.262
Caregiver’s social environment	Rural	1	ref.		
	Urban	1.258	0.501	3.157	0.625
Caregiver’s religious commitment	Strong	1	ref.		
	Moderate	0.902	0.342	2.378	0.835
	Indifferent or opposed	0.7	0.083	5.901	0.743
Caregiver’s education	Elementary	1	ref.		
	Secondary	0.115	0.02	0.657	0.015 *
	Higher	0.163	0.03	0.886	0.036 *
Admission of the child to the center	Urgent	1	ref.		
	Planned	0.518	0.113	2.369	0.397
Mode of admitting a child to the center	Outpatient clinic	1	ref.		
	Day ward	0.491	0.166	1.447	0.197
	Inpatient ward	0.289	0.089	0.939	0.039 *
Reason for admitting the child to the center	Chronic ailments	1	ref.		
	Ailments that occurred suddenly	1.666	0.467	5.944	0.431
Caregiver’s age	(years)	1	0.937	1.066	0.992
Daily number of infections	(thousands)	1.039	0.982	1.098	0.181
Total number of infections	(thousands)	1	1	1.001	0.497
Daily number of deaths		1.001	0.999	1.003	0.259
Total number of deaths	(thousands)	1.01	0.987	1.033	0.384
Total number of recoveries	(thousands)	1	1	1.001	0.479
Number of hospitalizations	(thousands)	1.005	0.957	1.056	0.839
Daily number of quarantined people	(thousands)	1.002	0.998	1.006	0.43
Season of stay at the center	Summer	1	ref.		
	Autumn	0.618	0.164	2.33	0.477
	Winter	1.432	0.495	4.137	0.508
	Spring	0.872	0.106	7.161	0.898
Coronavirus infection status	Unknown	1	ref.		
	Negative (in child and caregiver)	0.44	0.156	1.242	0.121

Multivariate logistic regression, OR—odds ratio, CI—confidence interval, *p*—statistical significance coefficient, *—statistically significant relationship (*p* < 0.05).

**Table 13 ijerph-20-04564-t013:** Predictors of coexistence of anxiety and depression: multivariate analysis.

Characteristic		Multivariate Model			
		OR	95% CI		*p*
Caregiver’s gender	Male	1	ref.		
	Female	1.306	0.315	5.414	0.713
	Non-binary person	-	-	-	-
Caregiver’s status	Caregiver of a child	1	ref.		
	Caregiver of a disabled child	0.443	0.146	1.349	0.152
Caregiver’s social environment	Rural	1	ref.		
	Urban	1.659	0.57	4.828	0.353
Caregiver’s religious commitment	Strong	1	ref.		
	Moderate	0.792	0.249	2.514	0.692
	Indifferent or opposed	0.672	0.067	6.704	0.735
Caregiver’s education	Elementary	1	ref.		
	Secondary	0.07	0.009	0.542	0.011 *
	Higher	0.072	0.009	0.561	0.012 *
Admission of the child to the center	Urgent	1	ref.		
	Planned	0.914	0.078	10.752	0.943
Mode of admitting a child to the center	Outpatient clinic	1	ref.		
	Day ward	0.45	0.129	1.573	0.211
	Inpatient ward	0.224	0.058	0.855	0.029 *
Reason for admitting the child to the center	Chronic ailments	1	ref.		
	Ailments that occurred suddenly	1.593	0.226	11.246	0.64
Caregiver’s age	(years)	0.99	0.922	1.063	0.778
Daily number of infections	(thousands)	0.749	0.456	1.231	0.254
Total number of infections	(thousands)	0.977	0.926	1.03	0.388
Daily number of deaths		1.016	1	1.033	0.055
Total number of deaths	(thousands)	9.221	0.407	208.826	0.163
Total number of recoveries	(thousands)	0.972	0.927	1.018	0.224
Number of hospitalizations	(thousands)	0.682	0.142	3.277	0.632
Daily number of quarantined people	(thousands)	1.04	0.995	1.086	0.083
Season of stay at the center	Summer	1	ref.		
	Autumn	0.088	0.004	1.938	0.124
	Winter	0	0	148.467	0.157
	Spring	0	0	814.528	0.227

Multivariate logistic regression, OR—odds ratio, CI—confidence interval, *p*—statistical significance coefficient, *—statistically significant relationship (*p* < 0.05).

**Table 14 ijerph-20-04564-t014:** Predictors of comorbidity of anxiety and depression: multivariate analysis (two variables).

Characteristic		OR	95% CI		*p*
Caregiver’s education	Elementary	1	ref.		
	Secondary	0.094	0.016	0.557	0.009 *
	Higher	0.115	0.02	0.67	0.016 *
Mode of admitting a child to the center	Outpatient clinic	1	ref.		
	Day ward	0.418	0.136	1.285	0.128
	Inpatient ward	0.261	0.078	0.878	0.03 *

Multivariate logistic regression, OR—odds ratio, CI—confidence interval, *p*—statistical significance coefficient, *—statistically significant relationship (*p* < 0.05).

## Data Availability

The data are available from the correspondent author.

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
