# Peer review of "Impact of the COVID-19 Pandemic on the Level of Anxiety and Depression in Caregivers of Children Benefiting from Neurorehabilitation Services"

_ijerph, 2023, doi:10.3390/ijerph20054564_

Round 1
Reviewer 1 Report
This study explored which of the selected data presented in the media reflecting the intensity of the Covid-19 pandemic differentiate the level of anxiety and depression in caregivers of children neurorehabilitation services. They found that there was no statistically relationship between the data presented in the media about the intensity of the Covid-19 and the level of anxiety and depression of the studied caregivers.
Some comments are noted below.
1: Not all tables were mentioned in the manuscript, such as Table 1 and Table 14. Please check.
2: On page4 lines 139 and 140, “Parents were asked whether they listened to or read news about the severity of the Covid-19 pandemic in the media the day before completing the survey.” The percentage of subjects who listened to or read news about the severity should be calculated and reported. The person who paid close attention on this information may be influenced by the Covid-19 epidemic, please analysis this subgroup of subjects.
3: The periods of time of the data in Table 2 should be described clearly.
4: The annotation is needed of Q1 Q3 in Table3.
5: The information about comorbidity of depression and anxiety should add to the Table 4.
6: The research results show that education as an important predictor for both anxiety and depression, please discuss this result at length.
7: The interpretation about the findings are thin, please discuss the result of there was no statistically relationship between the intensity of the Covid-19 and the level of anxiety and depression of the studied caregivers in detail.
8: The female gender is one of the predictors of anxiety; this result may be caused by data imbalance, as showed in Table 1, 83.04% of caregiver’s gender. How to eliminate the impact of data imbalance?
9: In Discussion, on page 16, the content is repetitive in lines 401 to 405 and lines 423 to 427. Please check if there are any mistakes.
10: Why authors mentioned about “spinal muscular atrophy” on page 16 line 419? Does it have relationship with this study? Please describe clearly.
11. It is recommended to reorganize the discussion section.
Author Response
On behalf of the authors, I would like to thank the Reviewer 1 for all comments and suggestions for changes to the article. The implementation of these changes will undoubtedly increase the value of the reviewed text. Some of the changes we've made are quite extensive. In this way, we wanted to take into account all the reviewer's suggestions. All changes in the text of the article are marked in red. Below we present them in the order corresponding to the reviewer's comments. Please check the attachment.

Reviewer 2 Report
Dear authors,
I would like to thank you for the opportunity to review this manuscript. Here are my comments:
In this paper, the authors evaluate the level of anxiety and depression among the caregivers of children benefiting from neurorehabilitation services - in the inpatient ward, outpatient clinic and day ward of Neurological Rehabilitation of Children and Adolescents in Rzeszow, Poland - shortly after receiving some data reflected in the media about the intensity of the COVID-19 epidemic, in the period June 2020 - April 2021.
The main tool used in this study was the Hospital Anxiety and Depression Scale (HADS). The average intensity of anxiety (HADS) in the subjects was 6.37 points, the mean intensity of depression (HADS) was 4.09 points. From the group of 454 caregivers, there was a higher number of them who presented severe anxiety (16.08%) than those who presented severe depression (4.63%) during the COVID-19 epidemic. Clear symptoms of anxiety and depression coexisted in 4.19% of the cases.
The data presented in the media regarding the severity of the COVID infection at a given time did not correlate statistically significantly with the level of anxiety and depression among the caregivers of children benefiting from neurorehabilitation services.
A regression model parameter values shows a series of predictors of the level of anxiety (female sex, indifference or lack of religious commitment, admission to an inpatient ward) and the level of depression (secondary and primary education compared to higher education, child's ailments that occurred suddenly as the reason for admission to the hospital compared to those whose children were admitted to an outpatient department); in the case of the combination between anxiety and depression the predictors are: secondary and higher education of the caregiver compared to primary education and admission of a child to an inpatient ward compared to an outpatient department.
1. Introduction: I have no major remarks on the Introduction Section. Still, I suggest you to put the lines 62-69 near lines 83-88 and re-arrange them to a better understanding of the psychological impact of the conditions by the COVID-19 pandemic on children with neurological disorders.
2. Material and methods: I suggest you to make some minor changes. Reference [5] would be better to be moved to section Discussion.
3. Results: You could give-up to the lines 240-248, specifying only the statistically significant relationships (lines 249-257).
You could give up to the sub-chapter The predictors of the co-occurrence of anxiety and depression - univariate analysis because it has approximately the same results with the following sub-chapter which describes the multivariate analysis.
4. Discussion: Lines 351-366 could be added to Introduction or Conclusion section.
Paragraph 391-405 and paragraph 406-427 could be rewrite to explain the relevance of the predictors found in this study.
5. Conclusions: In this section I suggest you to show the most important aspects revealed by this study accordingly with the aim.
Author Response
On behalf of the authors, I would like to thank the Reviewer 2 for all comments and suggestions for changes to the article. The implementation of these changes will undoubtedly increase the value of the reviewed text. Some of the changes we've made are quite extensive. In this way, we wanted to take into account all the reviewer's suggestions. All changes in the text of the article are marked in red. Below we present them in the order corresponding to the reviewer's comments. Please check the attachment.

Round 2
Reviewer 1 Report
My concerns are well responded. Some minor concerns should be addressed.
1: In the Introduction section in line 64, please list the common types of disease included in neurological disorders.
2: Statistical significance results should be labeled in Table, please check.
Author Response
1: In the Introduction section in line 64, please list the common types of disease included in neurological disorders.
Response: I wrote one sentence in line 64 .
2: Statistical significance results should be labeled in Table, please check.
Response: Statistically significant dependencies have been marked with an asterisk in the tables, also I have unified abbreviations and explanations under the tables.
Additionally, I added 1 reference (my own) in the References section.
Once again, I would like to thank the Reviewer for his (or her) comments and suggestions which I find very helpful.